# Diagnosis and Management of Cancer Treatment-Related Cardiac Dysfunction and Heart Failure in Children

**DOI:** 10.3390/children10010149

**Published:** 2023-01-12

**Authors:** Mohamed Hegazy, Stephanie Ghaleb, Bibhuti B Das

**Affiliations:** 1University of Mississippi Medical Center Program, Jackson, MS 39216, USA; 2Division of Pediatric Cardiology, Department of Pediatrics, Children’s of Mississippi Heart Center, University of Mississippi Medical Center, Jackson, MS 39216, USA; 3Division of Pediatric Cardiology, Department of Pediatrics, McLane Children’s Baylor Scott and White Medical Center, Baylor College of Medicine-Temple, Temple, TX 76502, USA

**Keywords:** cardio-oncology, cardiotoxicity, cancer-therapy related cardiotoxicity, oncological cardiomyopathy, chemotherapy-induced heart failure in children, childhood cancer survivors, cancer in children

## Abstract

It is disheartening for parents to discover that their children have long-term cardiac dysfunction after being cured of life-threatening childhood cancers. As the number of childhood cancer survivors increases, early and late oncology-therapy-related cardiovascular complications continues to rise. It is essential to understand that cardiotoxicity in childhood cancer survivors is persistent and progressive. A child’s cancer experience extends throughout his lifetime, and ongoing care for long-term survivors is recognized as an essential part of the cancer care continuum. Initially, there was a lack of recognition of late cardiotoxicities related to cancer therapy. About 38 years ago, in 1984, pioneers like Dr. Lipshultz and others published anecdotal case reports of late cardiotoxicities in children and adolescents exposed to chemotherapy, including some who ended up with heart transplantation. At that time, cardiac tests for cancer survivors were denied by insurance companies because they did not meet appropriate use criteria. Since then, cardio-oncology has been an emerging field of cardiology that focuses on the early detection of cancer therapy-related cardiac dysfunction occurring during and after oncological treatment. The passionate pursuit of many healthcare professionals to make life better for childhood cancer survivors led to more than 10,000 peer-reviewed publications in the last 40 years. We synthesized the existing evidence-based practice and described our experiences in this review to share our current method of surveillance and management of cardiac dysfunction related to cancer therapy. This review aims to discuss the pathological basis of cancer therapy-related cardiac dysfunction and heart failure, how to stratify patients prone to cardiotoxicity by identifying modifiable risk factors, early detection of cardiac dysfunction, and prevention and management of heart failure during and after cancer therapy in children. We emphasize serial longitudinal follow-ups of childhood cancer survivors and targeted intervention for high-risk patients. We describe our experience with the new paradigm of cardio-oncology care, and collaboration between cardiologist and oncologist is needed to maximize cancer survival while minimizing late cardiotoxicity.

## 1. Introduction

More people are surviving cancer than ever in the United States (US) [1]. According to a new report from the American Association for Cancer Research, there are 18 million survivors (5.4% of the population) in the US as of January 2022, which is expected to increase to 26 million by 2040. Also, there was a 32% reduction in the US cancer death rate between 1991 and 2019. Many factors increase cancer survival, including decreasing tobacco use, improving early diagnosis, and new therapeutic approaches, particularly molecularly targeted therapies and immunotherapy [2]. Beyond the US, cancer is an ongoing global challenge, and there were an estimated 17.2 million new cancer cases and 10 million cancer deaths globally in 2019 [3]. Among children ages, one to 14, cancer is the second-leading cause of death, and the most diagnosed cancers are leukemia, lymphomas, and brain tumors [2]. Studies on cancer-therapy-related cardiac dysfunction (CTRCD) in children are limited compared to a vast number of research in adults. Many pediatric cancer therapy protocols are extrapolated from guidelines for adults. Given the differences in body composition and developmental changes in children, extrapolating adult outcome data to children is not always appropriate [4]. Approximately 10% of children treated with anthracyclines (Doxorubicin and Daunorubicin) at doses greater than 300 mg/m^2^ develop symptomatic cardiotoxicity, associated with high morbidity and mortality [5]. Childhood cancer survivors who received anthracyclines are 15 times more likely than the general population to have heart failure (HF) and eight times more likely to die from cardiovascular (CV) diseases [6]. CTRCD is a spectrum of cardiac dysfunction that ranges from asymptomatic cardiac injury detectable only by elevated biomarkers such as troponin or brain-type natriuretic peptide (BNP) to overt symptoms and clinical signs of HF [7,8]. Recent studies also reported associations between CV risk factors such as arterial hypertension, hyperlipidemia and existing myocardial diseases such as cardiomyopathy with late CTRCD, leading to an interest in cardiovascular risk factors before, during, and after chemotherapy, specific anthracyclines such as Doxorubicin and Daunorubicin [9].

Cardio-oncology is a rapidly evolving discipline that focuses on detecting, monitoring, and treating CV disease as a side effect of cancer therapy [10]. Although there is controversy about whether the emerging new healthcare field should be called cardio-oncology or onco-cardiology, it is a matter of semantics rather than substance. CTRCD will continue to increase as this cohort of the population expands in the future. Several organizations have established guidelines, specifically the National Comprehensive Cancer Network’s “Adolescents and Young Adults with Cancer” [11] and The Children’s Oncology Group’s “Long-Term Follow-Up Guidelines for Survivors of Childhood, Adolescent, and Young” [12], with detailed information for institutions interested in establishing and enhancing long-term follow-up programs for childhood cancer survivors. It is important to note that there are variations between centers regarding CV monitoring, which leads to substantial practice variability [13]. In this review, we discuss the pathological basis of CTRCD and HF, how to stratify patients prone to cardiotoxicity by identifying modifiable risk factors, early detection of cardiac dysfunction, and prevention and management of HF during and after cancer therapy in children. We emphasize serial longitudinal follow-ups of childhood cancer survivors and targeted intervention for high-risk patients. We describe our experience with the new paradigm of balancing the successful treatment of cancer with minimizing the incidence of late-onset CTRCD.

## 2. Pathophysiology of Cardiovascular Toxicity in Cancer Patients

### 2.1. Anthracyclines

The first reports of anthracyclines-related LV dysfunction and HF in the 1960s [14] marked a series of changes in oncology practices, including the dose limitation of anthracyclines and their analogs (Epirubicin, Idarubicin, and Mitoxantrone), assessment of cardiac function before initiating chemotherapy, and other strategies to reduce cardiotoxicity. In a landmark study in adults, Swain et al. reported a high incidence of Doxorubicin-related HF, and the incidence of HF is directly proportional to the total cumulative dose [15]. The Childhood Cancer Survivorship Study (CCSS) has demonstrated a higher risk of cardiotoxicity at lower cumulative doses of 250 mg/m^2^ rather than 400 mg/m^2^, as shown by Swain et al. [16]. CCSS has suggested that there may be no safe anthracycline dose among children. In children, there are no differences in cardiotoxicity outcomes between bolus and continuous infusion anthracyclines in one study, which is different from adults [17].

The exact mechanism of anthracycline-induced cardiotoxicity is not entirely understood, and the incidence of cardiomyopathy and HF in those treated with anthracyclines is variable. The most commonly used anthracycline in pediatric cancer therapy is Doxorubicin, which exerts its antineoplastic effect via topoisomerase IIα binding, dysregulates deoxyribonucleic acid in cardiomyocytes, and disrupts mitochondrial biogenesis and function. Doxorubicin induces lipid peroxidation at the cell and mitochondrial membranes by forming complexes with Fe^2+^ and inducing apoptosis, mitochondrial DNA damage, and energy depletion through its reactive oxygen species (ROS) production [18]. It is hypothesized that persistent mitochondrial damage in cardiomyocytes leads to late CTRCD. Anthracycline prevents the normal electron transfer chain in mitochondria. Instead of standard end products like water and adenosine triphosphate, free radical superoxide anions are produced excessively in the mitochondria of cardiomyocytes [19]. The generated free radicals act as secondary signaling molecules in various pathways involved in homeostasis, impair cell proliferation, and ultimately cause cell death [20]. Interestingly, however, using supplemental antioxidants to reverse anthracycline-induced cardiotoxicity has not been successful. Anthracycline can also cause an increase in mitochondrial iron-mediated ROS in cardiomyocytes [21]. Transgenic mice with upregulated mitochondrial iron exporters were found in a pre-clinical study to be protected from anthracycline cardiotoxicity. These findings have led to using Dexrazoxane, an iron chelator that reduces the mitochondrial iron-mediated generation of ROS and provides cardioprotection from anthracycline therapy without reducing its anticancer efficacy [22]. Dexrazoxane can allow safe escalation of the anthracycline dose and thus be more effective in treating cancer and, at the same time, can mitigate CTRCD.

### 2.2. Non-Anthracycline Agents

There is no safe cancer therapy, and any of the chemotherapeutic agents can lead to long-term cardiotoxicity [23]. Many non-anthracycline chemotherapy agents are thought to produce free radicals and induce inflammatory changes, though at much lower rates than their anthracycline counterparts. For example, studies have suggested alkylating agents such as Cisplatin and Carboplatin have the potential to both increase ROS production and interfere with the antioxidant system, causing cardiotoxicity [24]. Furthermore, Cisplatin and Carboplatin can induce late-onset CTRCD, so long-term cardiological follow-up is essential [25]. Cyclophosphamide has been associated with cardiomyocyte apoptosis, inflammation, endothelial dysfunction, calcium dysregulation, and mitochondrial damage, resulting in hypertension, cardiomyopathy, myocardial infarction, arrhythmias, and, ultimately, HF [26]. While cardiotoxicity from 5-Fluorouracil (5-FU) is rare, the latter can cause the acute coronary syndrome, which may be caused by a decrease in nitric oxide locally, causing coronary artery spasm and vasoconstriction, microvascular thrombus, and ultimately myocardial ischemia [27].

### 2.3. Radiation-Related Cardiotoxicity

According to the American Society of Clinical Oncology, cardiac complications develop in 10–30% of patients receiving radiation therapy (RT) [28]. Radiation first interacts with water molecules in the cell, produces ROS, and damages the mitochondria and DNA in myocytes, leading to myocardial fibrosis. It also causes endothelial dysfunction, resulting in intravascular thrombosis, myocardial ischemia, and interstitial fibrosis. Immediately following RT, pericarditis, and myocarditis may be observed; however, their incidence is low with newer linear accelerators, intensity modulation, real-time target verification, and 3-dimensional (3D) conformational RT compared to older 2-dimensional (2D) RT [29]. Radiotherapy to the heart during childhood is associated with progressive late cardiac findings such as cardiomyopathy, valvular heart disease, pericardial disease, intracardiac conduction delay, and premature coronary artery disease (CAD). Notably, the age at onset of late effects is variable and depends partly on the patient’s age when RT commenced. Cardiotoxicity is more common in patients who have previously received anthracycline-based chemotherapy and other underlying pre-existing cardiac risk factors. The total radiation dose and the volume of the heart exposed determine cardiotoxicity. It has been reported that the incidence of cardiotoxicity increases by 60% for every 1-Gy increase in mediastinal radiation dose [30]. The bottom line is that all attempts should be made to reduce the volume of the heart exposed to RT, especially while receiving RT to the mediastinum and thoracic spine area.

### 2.4. Targeted Cancer Therapies 

Targeted cancer therapies are increasingly used in pediatric cancer as our understanding of the mechanisms of cancer progression has improved. These therapies include immune checkpoint inhibitors or ICI (Ipilimumab, Nivolumab, Pembrolizumab, Atezolizumab, Avelumab, and Durvalumab), tyrosine kinase inhibitors or TKI (Imatinib, Dasatinib, Bevacizumab, Bosutinib, Sunitinib, Sorafenib, Pazopanib, Pronatinib, Nilotinib), human epidermal growth factor receptor 2 (HER2)-targeted therapy (Trastuzumab, Pertuzumab), and proteasome inhibitors (Bortezomib, Carfilzomib). Although there is great optimism for their role in revolutionizing cancer therapy, targeted cancer therapies, too, are associated with cardiotoxicity, among other complications. Cardiotoxicity is reported in 60% of chronic myeloid leukemia patients, of whom 190 patients under 20 years received TKI [31]. TKI (e.g., Trametinib, Cobimetinib) and vascular endothelial growth factor inhibitors (e.g., Sorafenib, Bevacizumab) have been linked with the development of LV dysfunction, HF, arrhythmias, QT interval prolongation, and arterial thrombosis. ICI stimulates T-cells by interfering with checkpoint molecules, leading to an enhanced antitumor immune response [32]. They can cause acute autoimmune myocarditis [33]. In addition, there is some evidence that vascular endothelial growth factor (VEGF) inhibitors cause cardiotoxicity by hampering angiogenesis and targeting hypoxia-inducible factor 1 alpha (HIF-1α), which leads to myocardial hypoxia and cardiac dysfunction [34]. Trastuzumab, an anti-human HER2 monoclonal antibody used to treat HER2-positive breast cancer and HER2-positive osteosarcoma in pediatric populations, did not significantly improve overall survival of osteosarcoma in children and was associated with the potential for cardiotoxicity [35].

### 2.5. Hematopoietic Stem Cell Transplantation and Cellular Therapy

Cardiotoxicity in children undergoing allogenic or autologous bone marrow transplantation (BMT) is determined by underlying cancer for which transplantation is required. When BMT is done after the remission of hematological cancers in children to remove the latent source of cancer cell production or to replenish damaged bone marrow pluripotent stem cells, they are at a higher risk of cardiotoxicity due to pre-treatment with high doses of anthracyclines [36]. Also, total body irradiation is often used to eliminate the cancer cells before BMT, thus increasing their risk for cardiotoxicity. Patients are often treated with immunosuppressants such as calcineurin inhibitors (e.g., Cyclosporin and Tacrolimus) and corticosteroids as prophylaxis for graft versus host disease (GVHD), a complication after BMT. Calcineurin inhibitors have been linked to systemic hypertension, diabetes, and renal failure [37]. 

Cellular therapy, such as chimeric antigen receptor—T (CAR-T) cells, is increasingly used for refractory blood cancers, especially acute lymphocytic leukemia (ALL). Patients treated with CAR-T cell therapy are at risk for cardiotoxicity (21%) because of treatments they may have received in the past (e.g., anthracyclines) and cytokine release syndrome (CRS) caused by the activated CAR-T cells and macrophages [38]. Shalabi et al. showed that supraphysiologic levels of cytokine release could lead to tachycardia, hypotension, troponin elevation, reduced LVEF, pulmonary edema, and cardiogenic shock in 12% of children after CAR-T cell therapy [39]. CAR-T cells cause direct cell-mediated cancer cell lysis and raise serum potassium and uric acid levels, resulting in tumor lysis syndrome. In children, Burstein et al. showed that 25% had hypotension and 10% had worsening cardiac function at six months following CAR-T cell therapy [40]. 

## 3. Oncological Cardiomyopathy and Heart Failure

The common pathophysiological abnormality in all CTRCDs is either systolic or diastolic cardiac dysfunction. Herrmann et al. [41] proposed that the decline in cardiac function can occur due to direct cardiomyocyte damage (anthracyclines, Cyclophosphamide, 5-FU, HER2 inhibitors, and RT), other secondary alterations such as microvascular damage and fibrosis leading to impairment of myocardial function (Cyclophosphamide, 5-FU, Cisplatin, and RT), or due to inflammatory myocarditis (immune checkpoint inhibitors and RT). CTRCD ultimately progresses to HF over time [8]. The National Cancer Institute proposes the common terminology criteria for adverse events due to cancer therapy in adults, which classify LV dysfunction and HF into grades 1 through 5. Grade 1 is asymptomatic elevations in biomarkers or abnormalities on imaging. Symptoms with mild and moderate exertions are classified as grades 2 and 3. Grade 4 includes severe, life-threatening symptoms requiring hemodynamic support, and grade 5 involves death [42]. One of the clinical definitions of cardiotoxicity in adults has been formulated by the cardiac review and evaluation committee supervising Trastuzumab clinical trials, which defined drug-associated cardiotoxicity as one or more of the following: (1) cardiomyopathy in terms of a reduction in LVEF, either global or more severe in the septum; (2) symptoms associated with HF; (3) signs associated with HF, such as S3 gallop, tachycardia, or both; (4) a reduction in LVEF from baseline that is in the range of less than or equal to 5% to less than 55% with accompanying signs or symptoms of HF or a reduction in LVEF in the range of equal to or greater than 10% to less than 55% without accompanying signs or symptoms [43]. These definitions include arbitrary LVEF cutoffs without considering baseline risk or pre-existing cardiac dysfunction and do not include subclinical cardiovascular damage that may occur early in response to some chemotherapeutic agents; thus, a normal LVEF does not exclude significant myocardial dysfunction.

Sixty-two percent of childhood cancer survivors have at least one chronic health condition, and 27% have a severe or life-threatening illness, such as a stroke, HF, or renal failure [44]. Cardiotoxicity due to anthracyclines is extensively studied in children. The current definition of high-dose anthracycline exposure within the Children’s Oncology Group is a Doxorubicin equivalent of 250 mg/m^2^. However, there are reports of children developing CV disease with doses as low as 60 mg/m^2^ [45]. A continuum of changes starting with increased cardiac biomarkers, regional myocardial deformation (abnormal strain), and asymptomatic LV systolic or diastolic dysfunction occurs during chemotherapy, ultimately leading to HF [46]. Cardiotoxicity can develop in a subacute, acute, or chronic manner. In children, acute or subacute cardiotoxicity is characterized by either abnormality in the conduction system in the heart, ventricular repolarization and electrocardiographic QT-interval changes, supraventricular and ventricular arrhythmias, or pericarditis and/or myocarditis-like syndromes, observed any time from the initiation of therapy up to two weeks after the termination of treatment. In childhood cancer survivors, chronic cardiotoxicity may be differentiated into two subtypes based on the onset of clinical symptoms. The first subtype occurs early, within one year after administration of chemotherapy, while the second subtype occurs late, more than one year after chemotherapy, and is generally progressive. Children treated with Doxorubicin have decreased LV wall thickness relative to somatic growth compared to healthy children at a median follow-up of 11.8 years [47]. Cardiac abnormalities are persistent and progressive after Doxorubicin therapy because damaged myocytes are not recovered, which are replaced by interstitial fibrosis [48]. Typically, an adult number of myocytes are present before the first year of life, and subsequent myocardial growth occurs by increasing the size of the cells. Therefore, damage to or loss of these cardiomyocytes might impair the heart’s ability to generate an average adult myocardial mass. Inadequate ventricular mass due to the thinning of cardiomyocytes results in reduced cardiac output due to restrictive physiology [49]. Thus, cardiotoxicity due to anthracyclines in children differs from that in adults. Cardiotoxicity most commonly manifests in adults as acquired dilated cardiomyopathy or ischemic cardiomyopathy. In contrast, in children, cardiac changes initially appear to have dilated cardiomyopathy, which seems to respond to HF therapy if started early, but, in some cases, remodeling of LV progresses to restrictive-type cardiomyopathy in later years [50]. Progressive restrictive cardiomyopathy is usually persistent because of a progressive fall in LV mass and cavity size compared to body size, which becomes a long-term risk factor for HF with preserved EF in childhood cancer survivors [51].

## 4. Diagnosis and Risk Stratification

Identifying high-risk patients and diagnosing cardiotoxicity early in children before symptoms of HF occur requires a careful assessment of CV risk factors before, during, and after cancer therapy with standard history, physical examination, clinical and laboratory tests for cardiac biomarkers, and imaging methods. The risk of cardiotoxicity depends on the patient’s cancer diagnosis, the treatment received, and adverse health behaviors such as tobacco smoking, drinking alcohol, drug use (e.g., cocaine, diet pills, ephedra, mahuang), poor dietary habits, sedentary lifestyles and comorbidities such as pre-existing cardiomyopathy and underlying congenital heart disease, hypertension, hyperlipidemia, diabetes, and obesity [52]. The Childhood Cancer Survivor Study and other studies identified factors that increase the risk of developing cardiac toxicity, including younger patient age (<5 years), African American race, female sex, trisomy 21, total anthracycline dose, concomitant radiation exposure, underlying heart disease, pre-modern radiation protocols (before 1975), and time since treatment [30]. Furthermore, there is evidence that subclinical cardiac damage in cancer patients exists even before cancer therapy, which emphasizes the need to evaluate cardiac status in patients before cancer therapy [53]. Most of the existing guidelines regarding monitoring for the development of CTRCD are established for adult patients, with limited discussion of adult survivors of pediatric cancer [8,13,54]. Pediatric patients who receive anthracyclines >400 mg/m^2^ will develop restrictive cardiac dysfunction 30 to 40 years after cancer therapy. This long gap is an essential drawback in following these patients for late CTRCD because most patients attain adulthood. There is no continuity of care from pediatric to adult life, thinking that there are normal, and they lose contact with their pediatricians and pediatric oncologists. The general principle of diagnosing baseline cardiovascular risks and early diagnosis of CTRCD during and following cancer therapy includes serum cardiac biomarkers and multi-modality imaging, as summarized in Figure 1.

### 4.1. Cardiac Biomarkers

Measuring serum cardiac biomarkers can be a useful diagnostic tool for baseline assessment, the diagnosis of sub-clinical CTRCD during and following treatment, identifying high-risk patients who benefit from cardioprotective therapy, and tailoring oncologic therapies to individual risk profiles [55]. Elevated cardiac troponin T and BNP or NT-pro-BNP levels identify subclinical cardiac injury in patients with cancer therapy without CAD and are associated with depressed LV function [56,57]. Myocardial injury (measured as serum cardiac troponin T ≥ 99th percentile) during anthracycline therapy is associated with lower LV mass, wall thickness, and echocardiographic remodeling five years later [58]. Also, abnormal NT-pro-BNP (cardiomyopathy, age > 1 yr abnormal if NT-pro-BNP ≥ 100 pg/mL; age < 1 yr abnormal if NT-pro-BNP ≥ 150 pg/mL) during the first 90 days of anthracycline therapy is significantly related to LV remodeling (thickness to dimension ratio) by echocardiogram four years later [58]. Thus, screening with serial biomarkers matters for secondary prevention and monitoring long-term CTRCD in childhood cancer survivors. Galectin-3 is a biomarker related to cardiac inflammation and fibrosis, whereas ST2 (also known as soluble interleukin-1 receptor-like 1) is elevated in cardiac remodeling [58,59]. In one study, elevations in myeloperoxidase have been shown to predict cardiotoxicity [60]. Recently, exosomes, a subgroup of extracellular vesicles modulating multiple pathophysiological processes, have been proven to be a valuable biomarker for Doxycycline-induced cardiotoxicity [61]. Standardizing routine biomarker use in this clinical setting for children receiving cancer therapy is urgently needed. Given the variability of chemotherapy schedules and the possible different release kinetics of various biomarkers, future research should clarify whether a multi-biomarker approach would permit better risk stratification for CTRCD.

### 4.2. Role of Echocardiography

#### 4.2.1. 2D Echocardiography

Echocardiography plays a pivotal role in detecting structural changes such as diminished LV contractility expressed as LVEF, reduced LV wall thickness, and progressive LV dilation (increased LV end-diastolic diameter) even without symptoms. Patients should be stratified as having stage B HF if any changes are noted by echocardiography, according to the most recent 2022 ACC/AHA guidelines [62]. Early detection and prompt therapy of cardiotoxicity appear crucial for substantial recovery of cardiac function [63]. In adult cancer survivors, an absolute reduction in LVEF > 10% from baseline is described as CTRCD by the European Society of Cardiology (ESC), American Society of Echocardiography (ASE), American Society of Cardiac-Oncology (ASCO), and European Society for Medical Oncology (ESMO) [8,64,65]. Baseline echocardiography is essential to rule out pre-existing myocardial dysfunction, and those with normal LVEF should have a serial assessment by echocardiography during and after the completion of chemotherapy. However, 2D-derived LVEF has numerous limitations, such as preload and afterload dependence. Because cancer patients receive varying amounts of intravenous fluid and constant changes in hemodynamic stressors occur, 2D-derived LVEF may not be sensitive or specific to predict risks for CTRCD [66]. Furthermore, LVEF measurement lacks reproducibility, and there is 10% inter- and intra-observer variability [67]. 

#### 4.2.2. 3D Echocardiography

3D echocardiography has been reported to have superior reproducibility compared with 2D LVEF and lower inter- and intra-observer variability [67]. 3D LVEF changes also precede 2D LVEF changes [68,69], and 3D-derived LVEF correlates better with LV function estimated by cardiac magnetic resonance (CMR) [70]. 3D echocardiography-derived LVEF is likely to improve the accuracy and reproducibility of estimated LV function [71]. Serial 3D echocardiographic measurements of LVEF may be an early marker to detect subtle changes in LV function and thus can determine if a cardioprotective agent should be used along with anthracycline therapy. Although preferable to 2D, 3D echocardiography is restricted by limited availability in all centers, increased cost, and the need for an experienced operator to obtain high-quality images in children.

#### 4.2.3. Speckle-Tracking Global Longitudinal Strain

The recent development of semi-automated myocardial strain imaging by speckle-tracking echocardiography (STE) provides a more sensitive and reproducible measurement of myocardial deformation by tracking speckle displacement during the cardiac cycle. Strain is expressed as a percentage corresponding with the deformation in a region of interest. Unlike the measurement of LVEF, the strain does not rely on volume overload, and strain imaging may permit early detection of subclinical cardiotoxicity [72]. Global longitudinal strain (GLS) is the most commonly used strain parameter to assess early regional LV function abnormalities and appears to be most suitable for monitoring serial changes [73]. According to ASE guidelines, GLS is determined as the average peak longitudinal strain of 17 LV segments from 3 standard apical views [74]. In adult cancer survivors, LV GLS falling below (−)18% (0% to −17.9%) or a > 15% relative decrease of this marker may suggest clinically significant cardiotoxicity [72]. Pre-treatment measurements of GLS in cancer patients were significantly lower in the CTRCD group than in the non-CTRCD group, which may indicate an increased baseline risk profile for cardiovascular disease. The study of Ali et al. supports this finding; their results demonstrated that reduced baseline GLS > −17.5% was a strong predictor of cardiac events in patients with hematologic cancers [75]. 

Studies have shown abnormal longitudinal and circumferential strain in anthracycline-treated childhood cancer survivors [76,77]. In adults, cardioprotective therapy guided by LV GLS is beneficial [78]. However, to date, no predictive data in children demonstrates that the early detection of cardiotoxicity by strain analysis will alter the clinical prognosis. Such data would facilitate the modification of chemotherapy and the introduction of therapy to minimize the impact of cardiotoxicity. Right ventricular (RV) strain parameters have recently been found to be a useful early marker for subclinical CTRCD [79]. Myocardial strain imaging is thus a promising clinical modality for early cardiotoxicity detection and long-term surveillance of cancer patients [80,81].

#### 4.2.4. Tissue Doppler Imaging

Tissue Doppler imaging (TDI) or tissue velocity imaging has been used to evaluate diastolic dysfunction in children [51]. Rajapreyar et al. showed that diastolic dysfunction could be detected by TDI earlier than systolic dysfunction in children treated with anthracyclines [82]. However, the predictive value of LV diastolic impairment to predict CTRCD is doubtful because of inconsistent results regarding its ability to predict the subsequent occurrence of systolic dysfunction [83]. Additional clinical research and more extensive trials are necessary to evaluate the prognostic role of TDI parameters in childhood cancer survivors.

#### 4.2.5. Role of the Stress Test and Stress-Echocardiography

Cardiopulmonary exercise testing (CPET) provides a simple, noninvasive method for assessing dyspnea by unmasking pathology that resting studies cannot elicit. In addition, it can screen for cardiac dysfunction and other causes of shortness of breath in patients receiving chemotherapy, such as pulmonary hypertension, evolving pneumonitis or fibrosis, mitochondrial myopathy, or deconditioning. For this reason, CPET studies may be the ideal first-line tool for the workup of subclinical cardiac dysfunction in childhood cancer survivors [84]. The role of stress echocardiography is potentially helpful in risk-stratifying patients undergoing cancer therapies associated with myocardial ischemia [85]. As such, dobutamine stress echocardiography is a sensitive method to detect subclinical and clinical cardiac dysfunction in long-term survivors of asymptomatic children treated with anthracycline chemotherapy [86,87]. 

### 4.3. Role of CMR

CMR is recommended in patients with poor-quality echocardiographic images and patients with pre-existing heart diseases (for example, hypertrophic or dilated cardiomyopathy) [88]. It has greater intra- and inter-observer reproducibility and may identify a higher prevalence of CTRCD compared to echocardiography [89]. CMR does not utilize ionizing radiation, has an excellent spatial resolution, and can provide detailed tissue characteristics, including myocardial edema, inflammation, and fibrosis, thus playing an essential role in the identification of early and late cardiotoxicity in patients after cancer therapy by use of late gadolinium enhancement (LGE) and quantitative mapping techniques (T1 and T2 mapping). CMR also accurately assesses RV function by quantifying RV end-diastolic and end-systolic volumes. Recent reports have demonstrated that CMR-derived LV strain allows the detection of subclinical LV dysfunction during and after potentially cardiotoxic cancer therapy [90]. A decreased LV mass index suggestive of myocardial atrophy or growth arrest is an independent predictor of cardiomyopathy in cancer patients treated with anthracyclines [91]. Also, children exposed to anthracyclines have been shown to have increased left atrial (LA) volume when measured by CMR [92]. The superior spatial resolution of CMR provides the additional benefit of obtaining LA strain throughout the cycle of LA emptying and offers excellent information on diastolic dysfunction in childhood cancer survivors [93].

### 4.4. Role of Cardiac Catheterization

Anticancer therapies can cause significant injury to the vasculature, resulting in angina, myocardial ischemia, stroke, arrhythmias, and HF, independently from the direct myocardial or pericardial damage that might occur. Moreover, cancer is generally associated with a hypercoagulable state, which increases the risk of acute thrombotic events. Consequently, the need for invasive evaluation and management in the cardiac catheterization laboratory for adult cancer patients has been growing [94]. The role of cardiac catheterization in children receiving cancer therapy is limited except where angina or CAD is suspected.

### 4.5. Role of Advanced Imaging CT/SPECT/PET

The role of computed tomography (CT) is limited in pediatric cardio-oncology patients. CT coronary angiography may serve as an alternative imaging modality to stress echocardiography in a baseline assessment, though its higher cost and radiation exposure may limit its widespread use [95]. A common cardiovascular complication of cancer therapies is impairment of coronary circulation, either through direct vascular damage or accelerated atherosclerosis [37,44]. Noninvasive methods for evaluating myocardial perfusion with such parameters as myocardial blood flow and coronary flow reserve quantification are desirable in cardio-oncology care. For years, single-photon electron computed tomography (SPECT) imaging has been one of the principal methods for evaluating flow-limiting coronary stenosis in cardio-oncology patients, with the most commonly used radiotracers being 99mTc-labeled Sestamibi and Tetrofosmin. A large adult study that monitored cardiac function using serial SPECT over seven years and involved nearly 1500 patients who received cumulative doxorubicin doses of ≥450 mg/m^2^ showed that 19% of patients were at high risk of cardiotoxicity (defined as LVEF < 50%, a drop in LVEF by ≥10%) [96]. Although monitoring resting LVEF by SPECT helps detect early anthracycline cardiotoxicity, it has a low sensitivity compared to advanced imaging such as positron emission tomography (PET), especially with vectors labeled with positron-emitting radionuclides (e.g., carbon-11, fluorine-18, gallium-68) [97]. Cardiac PET is the current gold standard to assess myocardial perfusion because of its higher spatiotemporal resolution, count sensitivity, and accuracy. It is also valuable for diagnosing coronary microvascular dysfunction, especially with chemotherapy like Cyclophosphamide and 5-FU in adults [98]. Furthermore, PET allows for the evaluation of myocardial viability [99]. Despite these advancements, a lack of validation in the pediatric population has limited the use of PET imaging techniques in pediatric cardio-oncology patients. Limited data regarding optimal thresholds to distinguish pathologic from normal hyperemic myocardial blood flow and coronary flow reserve are available in children. 

## 5. Management

It is important to note that there is a variation in the definition of CTRCD across national societal position statements and oncology trials in adults [42,43,54,55]. Also, rates of cardiotoxicity and HF among children differ from those in adults with different anticancer agents. Hence, a standard recommendation will not fit children and adults to guide prevention, monitoring, and treatment strategies. Nonetheless, a multidisciplinary team, including cardiologists, oncologists, and allied healthcare professionals, must consider multifactorial risk factors for each patient receiving cancer therapy. In particular, the pediatric cardiologist should inform the oncologist of the patient’s cardiovascular risk factors, pre-existing cardiac disease status, prognosis, and intended treatment plan. We summarize our general approach to screening and managing cardiovascular risk factors and CTRCD for most pediatric patients before, during, and after cancer therapy (Figure 2). 

### 5.1. Prevention

Prevention can be primary and applicable to all patients receiving cancer therapy with potential cardiotoxicity. Primary prevention aims to optimize pre-existing modifiable CV risk factors, including controlling high blood pressure, lowering cholesterol, maintaining a healthy blood glucose level, consuming a nutritious diet, and stopping smoking during and after cancer treatment. Moderate aerobic exercise in selected patients is a promising nonpharmacological strategy to decrease CTRCD [100]. A review of 56 studies involving 4826 participants showed improved quality of life and physical capacity during and after a physical training program [101]. Secondary prevention is identifying high-risk patients for the development of HF who show signs of early cardiac injury so that they can be monitored very closely and cardioprotective therapy can be initiated. [54]. Other secondary preventive strategies include reducing the anthracycline dose, administering anthracycline as a continuous infusion instead of a bolus, choosing liposomal formulations of anthracyclines, and using fewer cardiotoxic anthracycline analogs (Epirubicin, Idarubicin, and Mitoxantrone) [102]. Dexrazoxane is the only drug approved by the US Food and Drug Administration (FDA) for the secondary prevention of anthracycline-related cardiomyopathy. Dexrazoxane is an iron-chelating agent with documented cardioprotective effects [22,103]. Although it was initially thought that the cardioprotective effect of Dexrazoxane was related to its iron-chelating properties, leading to cytosolic iron sequestration, more recent evidence suggests that inhibition of Doxorubicin-topoisomerase complex formation, leading to a reduction of free radicals, may also play a role [104]. Dexrazoxane decreases the cardiotoxic effects of anthracyclines without reducing their anticancer efficacy [105]. Unlike in adults, where Dexrazoxane is delayed until the anthracycline dose is 300 mg/m^2^, in children, it is recommended to give Dexrazoxane with the first anthracycline dose to be effective and to minimize cardiotoxicity in high-risk patients. Cardiomyopathy surveillance in cancer survivors for late CTRCD is recommended in all patients who received a total anthracycline dose of ≥250 mg/m^2^ or an RT dose ≥ 35 Gy [13]. Cardiomyopathy surveillance in cancer survivors is reasonable if treated with a moderate dose of anthracycline (>100 mg/m^2^ and <250 mg/m^2^) or RT (>15 Gy to <35 Gy) [13]. Cardiomyopathy surveillance may be reasonable for survivors who received a total anthracycline dose <100 mg/m^2^ [13]. An echocardiogram is a primary modality for surveillance to determine LVEF, either by 2D or 3D echocardiography [13]. Strain analysis to determine GLS is recommended but not widely used in children yet. CMR and other advanced modes of imaging are indicated only in selected patients. We recommend cardiomyopathy surveillance for high-risk patients to begin no later than one year after chemotherapy and repeat, depending on each patient’s status. Suppose the patient is completely asymptomatic and the echocardiographic parameters are normal. In that case, it is recommended to repeat the echocardiogram two to five years after the completion of chemotherapy and every five years. 

The rationale for using standard HF medications such as angiotensin-converting enzyme (ACE) inhibitors and β-blockers is primarily extrapolated from the adult experience with limited pediatric data [62,106]. In 2006, Cardinale et al. demonstrated that ACE inhibitors and β-blockers were beneficial in adults with anthracycline-related cardiomyopathy if therapy is initiated soon after diagnosing LV dysfunction [63]. In another randomized trial to study the long-term cardiotoxicity of childhood cancer survivors, asymptomatic cancer survivors with preserved EF received either Enalapril or placebo [107]. The study found that patients treated with high-dose (300 mg/m^2^) anthracyclines derived the most benefit from Enalapril therapy—six out of seven cardiac events occurred in the placebo arm, and nearly all were among those treated with high-dose anthracyclines. As a result, while using Enalapril to mitigate or prevent the cardiotoxic effects of cancer therapy may appear intuitive [108,109,110], the long-term impacts of containing HF have been disappointing. A multicenter, double-blind, randomized trial (NCT02717507) is currently ongoing to evaluate the long-term efficacy of Carvedilol in preventing cardiomyopathy and/or HF in high-risk childhood cancer survivors exposed to high-dose anthracyclines [111]. The study results will be available after three years of follow-up. The result will provide much-needed information regarding pharmacological risk-reduction strategies for childhood cancer survivors at high risk for developing anthracycline-related HF.

Exercise is beneficial in certain groups of individuals but could be detrimental to a subgroup of survivors with restrictive physiology because unsupervised exercise can put them at risk for pulmonary congestion and arrhythmia [112]. For patient safety, individual exercise prescriptions based on the late effects of the individual patient should be developed rather than a group recommendation for all cancer survivors. They should be individually reevaluated over time since survivor health changes over time. Appropriate and safe increases in physical activity will decrease the survivor’s cardiovascular risks, improve mental health, and decrease adverse cardiometabolic effects. Statins may be cardioprotective by reducing oxidative stress and inflammation in patients with other cardiovascular risk factors, but the benefits and risks remain unclear in the absence of any long-term studies in childhood cancer survivors [113].

### 5.2. Treatment of HF

Children who develop HF during therapy or over the long term after cancer therapy should be treated according to guideline-directed medical treatment [114]. Starting therapy early after the development of ventricular dysfunction can improve systolic function in most patients [115]. Currently, a “multi-hit model,” i.e., combination HF therapies, is preferred for treating HF [116]. Complex neurohormonal activation may occur as a response to myocardial injury and correlate with the severity of HF. These observations form the rationale for neurohormonal antagonists for treating HF with beta-adrenergic receptor blockers, ACE inhibitors, angiotensin receptor blockers, and mineralocorticoid receptor antagonists [114]. Unfortunately, Enalapril improved LV systolic function over six years but deteriorated 6 and 10 years after treatment [58]. This is because the primary defect was LV wall thinning, which continues to deteriorate, and thus the short-term improvement is mainly related to lowered diastolic blood pressure. As such, no long-term study has shown any beneficial effects of ACE inhibitors in childhood cancer survivors on improving quality of life, providing long-term benefits, or reducing progression to HF or death. Sacubitril and Valsartan combination therapy have improved LV function and cardiac biomarkers in adults with long-standing cardiotoxicity-induced HF [117,118]. Recently, newer HF therapies, such as sodium-glucose co-transporter-2 (SGLT2) inhibitors, have improved outcomes in adults with diabetes mellitus and cancers treated with anthracyclines [119]. Based on therapies of proven benefit in adult HF studies, these newer HF agents may be helpful in children with cancer therapy-induced refractory HF [116]. Refractory pediatric HF patients may need hospitalization and inotropic therapy. In some patients with end-stage HF, advanced HF therapies such as mechanical circulatory support (MCS) and heart transplantation can be successfully implemented [120]. Patients can be supported with MCS while undergoing cancer therapy and ultimately be bridged to heart transplantation [121,122]. However, a recent single institutional study reported a higher incidence of RV failure (4 out of 6 patients) while supported by a left ventricular assist device in children with cancer therapy-induced advanced HF [122]. 

## 6. Conclusions

Cardiotoxicity associated with cancer therapeutics can be pervasive, persistent, and progressive and can be missed clinically. One of the major theories is that ongoing mitochondrial damage may be related to lifespan cardiotoxicity. Cardiovascular-related health burdens will increase as the number of cancer survivors grows. Cardiomyopathy and HF are leading causes of death in cancer survivors, especially in children, given the anticipated long post-cancer lifespan of these children. Genetics has long been postulated to be one of the reasons why some patients develop cardiotoxicity while others with the same risk factors do not [123]. Therefore, pharmacogenetic testing and individualized cancer therapy can be very effective while simultaneously limiting CTRCD in the future. Furthermore, utilization of multi-modality parameters, including serial monitoring of cardiac biomarkers, LVEF, and GLS, may help identify CTRCD in its early stages, improving clinical practice and benefiting patient care. In our experience, there is no single model for pediatric cardio-oncology care, but we believe that the relationship between physician and patient, shared decisions, and working with a multidisciplinary team will improve the care of childhood cancer survivors. As the childhood cancer survivor population ages, it is important to have longitudinal follow-ups, and learning from our patients will help us do better. The healthcare system in the US is in flux, relatively unstable, and politicized. It is a system dominated by crisis mode, with relatively meager resources allocated towards prevention. Childhood cancer survivors need lifelong monitoring, especially those high-risk for CTRCD. Advocacy for managing CTRCD is a critical issue, and interactions with policymakers, insurers, and funding institutions are essential for this to be a top priority.

## Figures and Tables

**Figure 1 children-10-00149-f001:**
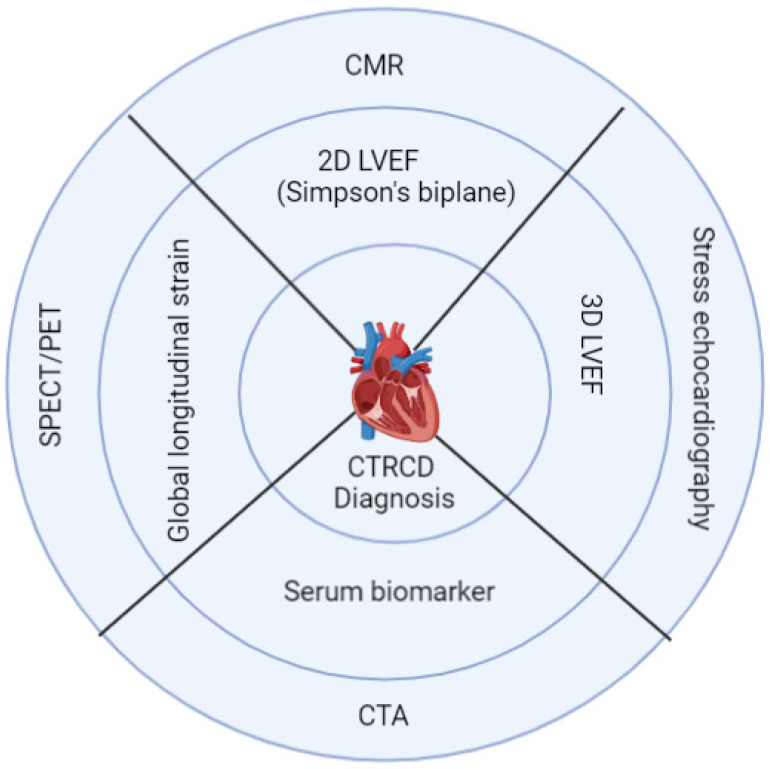
Serum cardiac biomarker and imaging to detect cardiotoxicity. First-line tests are represented in the inner circle, and second-line tests (represented in the outer circle), including advanced imaging techniques, are reserved for a certain group of patients where first-line tests are inadequate or specific information about coronary artery disease is needed. [2D: two-dimensional; LVEF: left ventricular ejection fraction; 3-D: three-dimensional; CTRCD: cancer treatment-related cardiac dysfunction; CMR: cardiac magnetic resonance imaging; CTA: computerized tomographic angiography; SPECT: single-photon electron computed tomography; PET: positron emission tomography].

**Figure 2 children-10-00149-f002:**
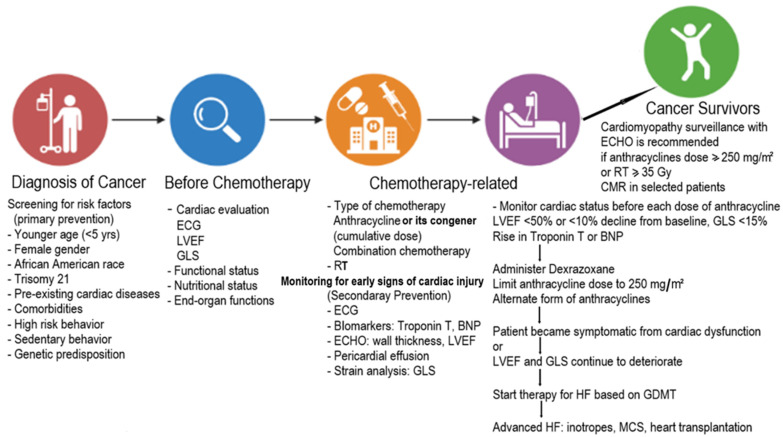
Continuum of cardiovascular care before, during, and after cancer therapy in children. [ECG: electrocardiogram; ECHO: echocardiogram; LVEF: left ventricular ejection fraction; GLS: global longitudinal strain; RT: radiation therapy; Gy: gray; GDMT: guideline-derived medical treatment; MCS: mechanical circulatory support; BNP: brain-type natriuretic peptide. Maximum anthracycline and radiation therapy cumulative doses used in our practice are based on references in the Journal of Clinical Oncology, 2017, 35, 893–911 [13].

## Data Availability

Not applicable.

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
