# Peer review of "Diagnosis and Management of Cancer Treatment-Related Cardiac Dysfunction and Heart Failure in Children"

_children, 2023, doi:10.3390/children10010149_

Round 1

Reviewer 1 Report

The authors have written a review report about diagnosis and management of cancer treatment-related cardiac dysfunction and heart failure with a focus on children as a group. The authors have elaborated on the cause of cardiac dysfunction due to different types of anti neoplastic therapies and about stratification of risk factors, various diagnosis techniques and disease biomarkers markers.

Comments -

1- Minor grammar and editing errors need to be corrected in the manuscript, (Line 96, Line 168, Line 233),

2- The prognosis and number of people affected with cardiac dysfunction by non-anthracycline agents, targeted cancer therapies and hematopoietic stem cell transplant therapies must be added just like they have been mentioned for anthracycline and radiation therapy.

3- Please add the abbreviations to the fig1 and in the figure legend, the figure quality and presentation must be improved, the figure legend in both figure 1 and 2 must be improved

4- Exosomes as biomarkers and for therapy have been fetching a huge attention in multiples diseases including cancer and cardiac dysfunction due to anthracyclines, following papers can be referred here

Potential of exosomes as diagnostic biomarkers and therapeutic carriers for doxorubicin-induced cardiotoxicity https://www.ncbi.nlm.nih.gov/pmc/articles/PMC8040474/

Inhibition of Glioma Cells' Proliferation by Doxorubicin-Loaded Exosomes via Microfluidics https://pubmed.ncbi.nlm.nih.gov/33149579/

Exosomes: Small vesicles with big roles in cancer, vaccine development, and therapeutics https://www.sciencedirect.com/science/article/pii/S2452199X21004059

5- Figure 2 needs to be changed, the presentation is poor and text is not clearly visible, the text could be reduced and presented well, also there is an editing error for the figure legend, please correct this.

Author Response

Response to Reviewer-1

The authors have written a review report about diagnosis and management of cancer treatment-related cardiac dysfunction and heart failure with a focus on children as a group. The authors have elaborated on the cause of cardiac dysfunction due to different types of anti neoplastic therapies and about stratification of risk factors, various diagnosis techniques and disease biomarkers markers.

Comments -

  • Minor grammar and editing errors need to be corrected in the manuscript, (Line 96, Line 168, Line 233)
  • We corrected the mistakes
  • The prognosis and number of people affected with cardiac dysfunction by non-anthracycline agents, targeted cancer therapies and hematopoietic stem cell transplant therapies must be added just like they have been mentioned for anthracycline and radiation therapy.
  • We reviewed PUBMED and Cochrane reviews. Most other cancer therapies focus on adults. We added where the numbers are available. Unfortunately, in childhood cancer survivors, we are still learning and prognosis for all cancer therapies are not available.

3- Please add the abbreviations to the fig1 and in the figure legend, the figure quality and presentation must be improved, the figure legend in both figure 1 and 2 must be improved

  • We added all abbreviations below Figure 1 and Figure 2.

4- Exosomes as biomarkers and for therapy have been fetching a huge attention in multiples diseases including cancer and cardiac dysfunction due to anthracyclines, following papers can be referred here

Potential of exosomes as diagnostic biomarkers and therapeutic carriers for doxorubicin-induced cardiotoxicity https://www.ncbi.nlm.nih.gov/pmc/articles/PMC8040474/

Inhibition of Glioma Cells' Proliferation by Doxorubicin-Loaded Exosomes via Microfluidics https://pubmed.ncbi.nlm.nih.gov/33149579/

Exosomes: Small vesicles with big roles in cancer, vaccine development, and therapeutics https://www.sciencedirect.com/science/article/pii/S2452199X21004059

  • Thank you. We reviewed these papers on exosomes and added (Reference 61) in the revised manuscript under the section “cardiac biomarkers” and reference.

5- Figure 2 needs to be changed, the presentation is poor and text is not clearly visible, the text could be reduced and presented well, also there is an editing error for the figure legend, please correct this.

  • We revised Figure -2 and edited the Figure 2 legend.

Reviewer 2 Report

Children (ISSN 2227-9067)

The following is an overview of the article Diagnosis and Management of Cancer Treatment-Related Cardiac Dysfunction and Heart Failure in Children (children-2133534). In this study, author(s) proposed to this review discusses the pathological basis of cancer-therapy-related cardiac dysfunction and heart failure, how to stratify risk factors for cardiotoxicity by identifying modifiable risk factors, early detection of cardiac dysfunction, and prevention and management of heart failure during and after cancer therapy in children. The manuscript has contributions to the area of Biomedical Diagnosis.

The author(s) stated in the first part of the study; more people are surviving cancer than ever in the United States (US). According to a new report from the American Association for Cancer Research, there are 18 million survivors (5.4% of the population) in the US as of January 2022, which is expected to increase to 26 million by 2040. Also, there was a 32% reduction in the US cancer death rate between 1991 and 2019. Many factors are responsible for increasing cancer survival including decreasing tobacco use, improving early diagnosis and new therapeutic approaches, particularly molecularly targeted therapies and immunotherapy. Beyond the US, cancer is an ongoing global challenge, and there were an estimated 17.2 million new cancer cases and 10 million cancer deaths globally in 2019. Among children ages one to 14, cancer is the second-leading cause of death, and the most diagnosed cancers are leukemia and brain tumors. Compared to the vast number of studies on cancer treatment related cardiac dysfunction (CTRCD) in adults, the number of similar studies in children is sparse. Hence, many pediatric cancer therapy protocols are extrapolated from those for adults, which is not always appropriate, given the differences in body composition and  developmental changes in children. It is estimated that approximately 10% of children  treated with anthracyclines (doxorubicin/daunorubicin) doses greater than 300 mg/m2 develop symptomatic cardiotoxicity, associated with high morbidity and mortality. It is disheartening for parents to discover that their children have long-term cardiac dysfunction after being cured of life-threatening childhood cancers. As the number of childhood cancer survivors are increasing, early and late oncology-therapy-related cardiovascular complications continue to rise. The cancer experience in a child extends throughout the lifetime, and ongoing care for long-term survivors is recognized as an essential part of the cancer care continuum. Cardio-oncology is an emerging field of cardiology that focuses on the early detection of cancer therapy-related cardiac dysfunction occurring before, during, and after oncological treatment. This review discusses the pathological basis of cancer-therapy-related cardiac dysfunction and heart failure, how to stratify risk factors for cardiotoxicity by identifying modifiable risk factors, early detection of cardiac dysfunction, and prevention and management of heart failure during and after cancer therapy in children.

The author(s) stated in the last part of the study; recently, it has been identified a possible relationship between the development of cardiomyopathy and anthracyclines therapy. Cardiomyopathy and HF are leading  causes of death in cancer survivors, especially in children, given the anticipated long post-507 cancer lifespan of these children. Genetics has long been postulated to be one of the reasons why some patients develop cardiotoxicity while others with the same risk factors do not. Therefore, pharmacogenetics testing and individualized cancer therapy can be very effective while simultaneously limiting CTRCD in future. Furthermore, utilization of multi-modality parameters, including serial monitoring of cardiac biomarkers, LVEF, and GLS, may help identify CTRCD in its early stages, improving clinical practice and benefiting patient care. Collaboration amongst specialties and across centers will provide critical data to further advance the rapidly growing field of cardio-oncology in the future.

However, some points must be highlighted so that the author(s) can review and submit in another round of review: The following corrections are considered to be beneficial for the strengthening of the article.

1. The Conclusions should be reviewed again. The original aspect of the study and its difference from other studies should be clearly explained. (The conclusion should be explored better and it needs to contemplate the eventual restrictions of the developed technique to address future works in this area.)

2. The abstract must be make strong. Abstract should be reviewed again.

3. It has been a comprehensive study in the literature in recent years. If there are more current literature studies, these should be examined in detail and added to the literature section (Especially, Heart Failure, Electrocardiography, Cardiac disease and Biomedical Diagnosis identification studies.). It is a suggestion for the literature part of the article to be more comprehensive. It may be useful to include relevant articles in 2018-2022 in references. As an example, I think it might be useful to add the article to references, such as the articles below, to keep the article updated as a literature. (3.1) A New Approach for Congestive Heart Failure and Arrhythmia Classification Using Angle Transformation with LSTM. Arabian Journal for Science and Engineering, 1-17. 3.2) A novel ECG diagnostic system for the detection of 13 different diseases. Engineering Applications of Artificial Intelligence, 107, 104536. 3.3) A new approach for congestive heart failure and arrhythmia classification using Down Sampling Local Binary Patterns with LSTM, Turkish Journal of Electrical Engineering and Computer Sciences, DOI: 10.3906/elk-2201-162. 3.4) The Use of Wearable ECG Devices in the Clinical Setting: a Review. Current Emergency and Hospital Medicine Reports, 1-6. 3.5) Multi-Layer Co-Occurrence Matrices for Person Identification from ECG Signals. Traitement du Signal, 39(2).)

4. The authors should compare the results of their method with those of previous studies. As mentioned in the literature, there are several methods with very high accuracy, even better than the proposed method. Author(s) can do compare table (A new table can add about previous studies to result section.).

5. The motivations of the proposed method are not clear. Which problem does the proposed method attempt to solve? Why the other existing diagnosis methods failed to solve it? What are the advantages of the proposed method compared to other methods? Those should be illustrated more clearly.

6. Carefully check all grammatical error. Still, the English language should be improved. I suggest asking for help from a native English.

7. Some sentences have spelling errors. (Punctuation marks, spaces, etc.). Some places should be left space.

I think it is ACCEPTABLE after the MAJOR Revisions mentioned.

Author Response

Response to Reviewer-2

The following is an overview of the article Diagnosis and Management of Cancer Treatment-Related Cardiac Dysfunction and Heart Failure in Children (children-2133534). In this study, author(s) proposed to this review discusses the pathological basis of cancer-therapy-related cardiac dysfunction and heart failure, how to stratify risk factors for cardiotoxicity by identifying modifiable risk factors, early detection of cardiac dysfunction, and prevention and management of heart failure during and after cancer therapy in children. The manuscript has contributions to the area of Biomedical Diagnosis.

The author(s) stated in the first part of the study; more people are surviving cancer than ever in the United States (US). According to a new report from the American Association for Cancer Research, there are 18 million survivors (5.4% of the population) in the US as of January 2022, which is expected to increase to 26 million by 2040. Also, there was a 32% reduction in the US cancer death rate between 1991 and 2019. Many factors are responsible for increasing cancer survival including decreasing tobacco use, improving early diagnosis and new therapeutic approaches, particularly molecularly targeted therapies and immunotherapy. Beyond the US, cancer is an ongoing global challenge, and there were an estimated 17.2 million new cancer cases and 10 million cancer deaths globally in 2019. Among children ages one to 14, cancer is the second-leading cause of death, and the most diagnosed cancers are leukemia and brain tumors. Compared to the vast number of studies on cancer treatment related cardiac dysfunction (CTRCD) in adults, the number of similar studies in children is sparse. Hence, many pediatric cancer therapy protocols are extrapolated from those for adults, which is not always appropriate, given the differences in body composition and  developmental changes in children. It is estimated that approximately 10% of children  treated with anthracyclines (doxorubicin/daunorubicin) doses greater than 300 mg/m2 develop symptomatic cardiotoxicity, associated with high morbidity and mortality. It is disheartening for parents to discover that their children have long-term cardiac dysfunction after being cured of life-threatening childhood cancers. As the number of childhood cancer survivors are increasing, early and late oncology-therapy-related cardiovascular complications continue to rise. The cancer experience in a child extends throughout the lifetime, and ongoing care for long-term survivors is recognized as an essential part of the cancer care continuum. Cardio-oncology is an emerging field of cardiology that focuses on the early detection of cancer therapy-related cardiac dysfunction occurring before, during, and after oncological treatment. This review discusses the pathological basis of cancer-therapy-related cardiac dysfunction and heart failure, how to stratify risk factors for cardiotoxicity by identifying modifiable risk factors, early detection of cardiac dysfunction, and prevention and management of heart failure during and after cancer therapy in children.

The author(s) stated in the last part of the study; recently, it has been identified a possible relationship between the development of cardiomyopathy and anthracyclines therapy. Cardiomyopathy and HF are leading  causes of death in cancer survivors, especially in children, given the anticipated long post-507 cancer lifespan of these children. Genetics has long been postulated to be one of the reasons why some patients develop cardiotoxicity while others with the same risk factors do not. Therefore, pharmacogenetics testing and individualized cancer therapy can be very effective while simultaneously limiting CTRCD in future. Furthermore, utilization of multi-modality parameters, including serial monitoring of cardiac biomarkers, LVEF, and GLS, may help identify CTRCD in its early stages, improving clinical practice and benefiting patient care. Collaboration amongst specialties and across centers will provide critical data to further advance the rapidly growing field of cardio-oncology in the future.

However, some points must be highlighted so that the author(s) can review and submit in another round of review: The following corrections are considered to be beneficial for the strengthening of the article.

  1. The Conclusions should be reviewed again. The original aspect of the study and its difference from other studies should be clearly explained. (The conclusion should be explored better and it needs to contemplate the eventual restrictions of the developed technique to address future works in this area.)

- We revised the conclusion section.

  1. The abstract must be make strong. Abstract should be reviewed again.

- We revised the abstract.

  1. It has been a comprehensive study in the literature in recent years. If there are more current literature studies, these should be examined in detail and added to the literature section (Especially, Heart Failure, Electrocardiography, Cardiac disease and Biomedical Diagnosis identification studies.). It is a suggestion for the literature part of the article to be more comprehensive. It may be useful to include relevant articles in 2018-2022 in references. As an example, I think it might be useful to add the article to references, such as the articles below, to keep the article updated as a literature. (3.1) A New Approach for Congestive Heart Failure and Arrhythmia Classification Using Angle Transformation with LSTM. Arabian Journal for Science and Engineering, 1-17. 3.2) A novel ECG diagnostic system for the detection of 13 different diseases. Engineering Applications of Artificial Intelligence, 107, 104536. 3.3) A new approach for congestive heart failure and arrhythmia classification using Down Sampling Local Binary Patterns with LSTM, Turkish Journal of Electrical Engineering and Computer Sciences, DOI: 10.3906/elk-2201-162. 3.4) The Use of Wearable ECG Devices in the Clinical Setting: a Review. Current Emergency and Hospital Medicine Reports, 1-6. 3.5) Multi-Layer Co-Occurrence Matrices for Person Identification from ECG Signals. Traitement du Signal, 39(2).)

- Thank you, Sir. We reviewed most of your suggestions on AI and its application in HF and arrhythmia. Pediatric cardio-oncology is still in the very initial stage and has a lot to evolve, and may be in future reviews, and we will have more data on childhood cancer survivors. We try to limit the review to pediatric cardiac toxicity only, and we want to share our experience.

  1. The authors should compare the results of their method with those of previous studies. As mentioned in the literature, there are several methods with very high accuracy, even better than the proposed method. Author(s) can do compare table (A new table can add about previous studies to result section.).

- This is a review article. We tried to review the articles published in children. As you know, the literature on adults is very vast. But, the evidence of CTRCD in children is still evolving. We summarized all data that are relevant to pediatrics.

  1. The motivations of the proposed method are not clear. Which problem does the proposed method attempt to solve? Why the other existing diagnosis methods failed to solve it? What are the advantages of the proposed method compared to other methods? Those should be illustrated more clearly.

- We have described our experience in taking care of cardio-oncology pediatric patients. There is a lot of variability in practice from center to center. We wanted to share our experience. Pediatric solid tumors are not as frequent as in adults. Most pediatric drugs are off-level use. Unfortunately, most off-level use in children is only for relapsing cancer. We propose how to take care of pediatric cardio-oncology patients in general.

  1. Carefully check all grammatical error. Still, the English language should be improved. I suggest asking for help from a native English.

- We did our best using the English language checking website provided by the Journal, which was also revised by English speaking independent person.

  1. Some sentences have spelling errors. (Punctuation marks, spaces, etc.). Some places should be left space.

- We corrected most errors.

  1. I think it is ACCEPTABLE after the MAJOR Revisions mentioned.

- Thank you. We hope our revised version will satisfy you and better serve pediatric cardio-oncology care.

Reviewer 3 Report

Children-2133534, Diagnosis and Management of Cancer Treatment-Related Cardiac Dysfunction and Heart Failure in Children, by Mohamed Hegazy et al. The authors aimed in their review to discuss the pathological basis of cancer-therapy-related cardiac dysfunction and heart failure, stratify risk factors for cardiotoxicity, and management of heart failure during and after cancer therapy in children.

Major:

1) Page 2; Pathophysiology of Cardiovascular Toxicity in Cancer Patients. Since the main scope/focus of the current review is children, is the mechanism(s) of cardiotoxicity due to cancer therapies like anthracyclines is(are) similar in both children and adults? Is there any literature discussing the anthracycline toxicity specifically in children with special emphasis on its mechanism? Please further explain/mention in the manuscript, citing the appropriate references, whenever possible.

2)  Page 4; lines 146-147: “Although there is great optimism for their role in revolutionizing cancer therapy, they, too, are associated with cardiotoxicity, among other complications”. Since this sentence is considered as a separate one, and “targeted cancer therapies” is not mentioned/spelled in the first part of the sentence, it is better to replace “they” by “targeted cancer therapies” for better understanding.

3) Page 5; lines 214-215: “62% of childhood cancer survivors have at least one chronic health condition, and 27% had a severe or life-threatening illness, such as stroke, HF, or renal failure”. It is not advisable to start a sentence with numeral, thus, please reword the sentence or spell out the number. Furthermore, it is recommended to keep the verb tense constant throughout the sentence. Accordingly, please replace “had” with “have”.

4) Page 5-6, lines 248-252. It is unclear whether the risk factors of cardiotoxicity are in adults or children or both. Please specify.

5) Page 9, lines 415-417: “Despite these advancements, several challenges have limited the use of these PET imaging techniques in pediatric cardio-oncology patients.” What are the challenges? Please elaborate more with references, whenever possible.

6)  There are some occasions where the authors are required to support their statements by prior literature/references, whenever possible.

a.  Page 2, lines 58-60: “however, a five to six-fold increase in.……cancer survivors”.

b.  Page 4, lines 156-159: “Trastuzumab, an anti-human HER2……was associated with potential for cardiotoxicity”.

c. Page 6, lines 278-279: “However, neither of these prior biomarkers.......patients treated with cancer therapy”.

d.    Page 9, lines 397-399: “A common cardiovascular complication of cancer therapies…...accelerated atherosclerosis”.

7)  Numbering; section 4 (Pages 5-9): it is unclear why the authors opted to use the formatting as 4.A, 4.B (4.B.1 through 4.B.5), 4.C etc. instead of using 4.1, 4.2 (4.2.1 through 4.2.5), 4.3 etc. The latter format of numbering is more appropriate to be consistent with the rest of numbering in the manuscript (recommended by reviewer).

8)    References: 

a. The reference no.16 (Kadan-Lottick NS et al.), for some reason, has been wrongly written/divided in the bibliography as two references (no. 16 and no. 17) instead of being listed as no. 16 only. This error shifted the rest of the bibliography by one reference. For example, Hermann et al study is listed as no. 42 in the bibliography, while indeed this reference is cited as reference no. 41 in the text (page 4, line 191), which is more accurate. Likewise, on page 8, lines 332-334, Ali et al study is cited as reference no. 75 in the text; however, the same reference is listed as no. 76 in the bibliography. The same pattern applies to all references from no. 18 till the end. Authors should correct the bibliography by fixing reference no. 16, so that the rest of the bibliography matches the in-text citation.

b. The authors must follow the journal guidelines in regard of reference style and most importantly make it consistent throughout the bibliography. It is noted that different styles have been presented in the bibliography of the current review, which need the attention of the authors. For examples: the authors’ names of certain references are separated by comma, while others are separated by semicolon (example: references no. 32, 33 and 36), whereas some references are shown to have no separation between authors names (example: references no. 18-22).

Minor:

1)  Please check for and separate the words that are erroneously sticked together in the manuscript. Here are some examples:

a.     Page 2, line 69: “asour” should be “as our”.

b.     Page 4, line 149: “131TKI” should be “131 TKI”.

c.     Page 4, line 167: “forBMT” should be “for BMT”.

d.     Page 5, line 229: “andis” should be “and is”.

e. Page 5, line 233: “insteadreplaced” should be “instead replaced”.

f.      Page 7, line 291: “recent2022” should be “recent 2022”.

g.     Page 9, line 413: “Itis” should be “It is”.

2) Page 2, Line 86: “(DNA), , including”. Please remove the extra comma.

3) Page 3, Line 127: “Immediately following RT, pericarditis, and myocarditis may be observed”. Please remove the comma after “pericarditis”.

4)   Page 10, line 436: “LV EF” should be written as one word “LVEF”.

5) There are some places in the manuscript where it seems there is extra space between words (it is unclear whether those spaces were originally in the submitted document or appeared in the generated PDF). The reviewer recommends the authors to double check and remove extra space, if found. Here are some examples:

a.  Page 3, line 96. “triphosphate,  free radical” between “comma” and “free”.

b.  Page 4, line 164. “children are  at a higher” between “are” and “at”.

c. Page 5, line 202. “involves death.” 42 One of” between “quotation mark” and “reference number”.

d.  Page 7, line 294. “described as  CTRCD by” between “as” and “CTRCD”.

e.   Page 7, line 309. “also preceded  2D LVEF” between “preceded” and “2D”.

f.    Page 8, line 341. “parameters have  also been” between “have” and “also”.

g.  Page 11, line 470. “or a placebo. 107” between “period” and “reference number”.

Author Response

Response to Reveiewer:3

Children-2133534, Diagnosis and Management of Cancer Treatment-Related Cardiac Dysfunction and Heart Failure in Children, by Mohamed Hegazy et al. The authors aimed in their review to discuss the pathological basis of cancer-therapy-related cardiac dysfunction and heart failure, stratify risk factors for cardiotoxicity, and management of heart failure during and after cancer therapy in children.

Major:

  • Page 2; Pathophysiology of Cardiovascular Toxicity in Cancer Patients. Since the main scope/focus of the current review is children, is the mechanism(s) of cardiotoxicity due to cancer therapies like anthracyclines is(are) similar in both children and adults? Is there any literature discussing the anthracycline toxicity specifically in children with special emphasis on its mechanism? Please further explain/mention in the manuscript, citing the appropriate references, whenever possible.
  • Thank you. We have written in the text that the mechanism of anthracycline toxicity is unknown. We wrote the differences in the adult study (Swain et al. vs. COG study) and emphasized the differences between children and adults. We also included a study that showed that anthracycline given continuous infusion does not make any difference in children, unlike adults. (Added Ref 17) This is highlighted in the text
  • Page 4; lines 146-147: “Although there is great optimism for their role in revolutionizing cancer therapy, they, too, are associated with cardiotoxicity, among other complications”. Since this sentence is considered as a separate one, and “targeted cancer therapies” is not mentioned/spelled in the first part of the sentence, it is better to replace “they” by “targeted cancer therapies” for better understanding.
  • We added this in the revised text (highlighted)
  • Page 5; lines 214-215: “62% of childhood cancer survivors have at least one chronic health condition, and 27% had a severe or life-threatening illness, such as stroke, HF, or renal failure”. It is not advisable to start a sentence with numeral, thus, please reword the sentence or spell out the number. Furthermore, it is recommended to keep the verb tense constant throughout the sentence. Accordingly, please replace “had” with “have”.
  • We changed the number to words and corrected the verb.
  • Page 5-6, lines 248-252. It is unclear whether the risk factors of cardiotoxicity are in adults or children or both. Please specify.
  • We added children and corrected the sentences.
  • Page 9, lines 415-417: “Despite these advancements, several challenges have limited the use of these PET imaging techniques in pediatric cardio-oncology patients.” What are the challenges? Please elaborate more with references, whenever possible.
  • Thank you. We added ”lack of validation in the pediatric population” as a cause of PET not being used in children.

6)  There are some occasions where the authors are required to support their statements by prior literature/references, whenever possible.

  1. Page 2, lines 58-60: “however, a five to six-fold increase in.……cancer survivors”.

- We edited and added a reference, “Childhood cancer survivors who received anthracyclines are 15 times more likely than the general population to have heart failure (HF) and eight times more likely to die from cardiovascular (CV) diseases.6

  1. Page 4, lines 156-159: “Trastuzumab, an anti-human HER2……was associated with potential for cardiotoxicity”.

- We added a new reference: 35

  1. Page 6, lines 278-279: “However, neither of these prior biomarkers.......patients treated with cancer therapy”.

- Thank you. This sentence is deleted as this contradicts the evidence written (prior 4 lines).

  1. Page 9, lines 397-399: “A common cardiovascular complication of cancer therapies…...accelerated atherosclerosis”

-Thank you. Added Reference 37 and 44

[FYI: Note: After changing everything you suggested, I wrote the lines in the revised manuscript. Then I got an email from the editorial staff that there were similarities as per authenticate report, so I changed and edited the manuscript. The line numbers changed again. I am sorry for this.]

Round 2

Reviewer 1 Report

Comments

1- The figure 2 is done unprofessionally, needs to be re-done, the text size is not uniform, some suggestions, made the headings bold , italic or underlined, the bullet points are having - in some areas and nothing in other areas, please make it uniform, the arrows are bold in some cases and of different size compared to others. Make use of illustrator , or photoshop, or inkscape or any other software to make the panels and text uniform, the abbreviations for BNP , GDMT are still missing from the figure legend

2- In the references section correct editing error in Ref.34

3- References list needs to be checked and re done, the eg. Ref 24 m in the manuscript there is no mention of carboplatin and or how cisplatin increases ROS production, it rather mentions some other mechanism for cardiotoxicity. Ref 32 does not have anything in the manuscript about autoimmune myocarditis.  Please correct all the references where ever necessary.

4- Please correct the line spacing in the author contributions

Author Response

Response to Reveiewer:1

  • The figure 2 is done unprofessionally, needs to be re-done, the text size is not uniform, some suggestions, made the headings bold , italic or underlined, the bullet points are having - in some areas and nothing in other areas, please make it uniform, the arrows are bold in some cases and of different size compared to others. Make use of illustrator , or photoshop, or inkscape or any other software to make the panels and text uniform, the abbreviations for BNP , GDMT are still missing from the figure legend
  • We revised the Figure.
  • In the references section correct editing error in Ref.34
  • It was a setting error, corrected.
  • References list needs to be checked and re done, the eg. Ref 24 m in the manuscript there is no mention of carboplatin and or how cisplatin increases ROS production, it rather mentions some other mechanism for cardiotoxicity. Ref 32 does not have anything in the manuscript about autoimmune myocarditis. Please correct all the references where ever necessary.
  • Ref 24 has discussed all non-anthracycline-induced cardiotoxicity, and it describes also carboplatin and cisplatin mechanisms of action. Ref 32 was misplaced; references are corrected.
  • Please correct the line spacing in the author contributions
  • The journal will reset; this is at the proof stage only. Thank you.

Reviewer 2 Report

Children (ISSN 2227-9067) | An Open Access Journal from MDPI

Dear Editor;

The author(s) made all the corrections mentioned (children-2133534 - Diagnosis and Management of Cancer Treatment-Related Cardiac Dysfunction and Heart Failure in Children).

The length of the paper is enough in terms of a scientific paper. Considering studies conducted and results obtained.

Author Response

Response to Reveiewer:2

Dear Editor;

The author(s) made all the corrections mentioned (children-2133534 - Diagnosis and Management of Cancer Treatment-Related Cardiac Dysfunction and Heart Failure in Children).The length of the paper is enough in terms of a scientific paper. Considering studies conducted and results obtained.

  • Thank you.